

# Identification of a metabolic-related gene signature predicting the overall survival for patients with stomach adenocarcinoma

Yuan Nie[*], Linxiang Liu[*], Qi Liu and Xuan Zhu

Department of Gastroenterology, First Affiliated Hospital of Nanchang University, Nan Chang, China
[*] These authors contributed equally to this work.

## ABSTRACT

**Background**. The reprogramming of energy metabolism and consistently altered metabolic genes are new features of cancer, and their prognostic roles remain to be further studied in stomach adenocarcinoma (STAD).

**Methods**. Messenger RNA (mRNA) expression profiles and clinicopathological data were downloaded from The Cancer Genome Atlas (TCGA) and the GSE84437 databases from the Gene Expression Omnibus (GEO) database. A univariate Cox regression analysis and the least absolute shrinkage and selection operator (LASSO) Cox regression model established a novel metabolic signature based on TCGA. The area under the receiver operating characteristic (ROC) curve (AUROC) and a nomogram were calculated to assess the predictive accuracy.

**Results**. A novel metabolic-related signature (including acylphosphatase 1, RNA polymerase I subunit A, retinol dehydrogenase 12, 5-oxoprolinase, ATP-hydrolyzing, malic enzyme 1, nicotinamide N-methyltransferase, gamma-glutamyl transferase 5, deoxycytidine kinase, galactosidase alpha, DNA polymerase delta 3, glutathione S-transferase alpha 2, N-acyl sphingosine amidohydrolase 1, and N-acyl sphingosine amidohydrolase 1) was identified. In both TCGA and GSE84437, patients in the high-risk group showed significantly poorer survival than the patients in the low-risk group. A good predictive value was shown by the AUROC and nomogram. Furthermore, gene set enrichment analyses (GSEAs) revealed several significantly enriched pathways, which may help in explaining the underlying mechanisms.

**Conclusions**. A novel robust metabolic-related signature for STAD prognosis prediction was conducted. The signature may reflect the dysregulated metabolic microenvironment and can provided potential biomarkers for metabolic therapy in STAD.

# INTRODUCTION

Stomach adenocarcinoma (STAD) is one of the five most common cancers and ranks third among the cancer-related deaths worldwide. Globally, there are approximately 951,600 new STAD cases and approximately 723,100 STAD-related deaths each year. STAD is harmful to human health and social development. Due to the lack of specific symptoms in the

Corresponding author
Xuan Zhu, waiyongtg@163.com

early stage of STAD, most patients have reached the middle and late stages of the disease when they are diagnosed, which then leads to a poor prognosis. According to the statistics, the 5-year survival rate of patients with advanced STAD is less than 20%, whereas the 5-year survival rate of patients with early diagnosis and surgical resection can increase to reach more than 90% (*Thrumurthy et al., 2015*; *Thrumurthy et al., 2013*). Therefore, early diagnoses and timely interventions are of great significance for improving the prognosis of patients with STAD. At present, the diagnosis of STAD is mainly based on the pathological examination using endoscopy and tissue biopsy, but this method is traumatic and costly and has a low patient compliance rate. Although common gastrointestinal tumor markers, such as CEA, CA72-4, and CA19-9, have been widely used in clinical practice, the positive rate of early diagnosis is limited and cannot meet the requirements of early screenings of STAD. Therefore, it is of great practical significance to develop noninvasive, sensitive and specific biomarkers for the diagnosis and prognosis of gastric cancer.

Metabolic disorder is a key event in cancer, and tumor-related metabolic changes are involved in the generation, maintenance and progression of tumors (*Beger, 2013*). When compared with that in normal tissues, there is a large amount of metabolic heterogeneity in cancer cells and tumors; this includes glucose and amino acids exhibiting imbalanced levels, which increases the demand for nitrogen (*Pavlova & Thompson, 2016*). In the occurrence and development of STAD, abnormal glycolysis and amino acid metabolism constitute the essence of metabolic phenotype changes in STAD (*Gu et al., 2016*). The regulation of tumor metabolism mainly involves the activation of oncogenes, the inactivation of tumor suppressor genes and the changes in metabolic pathways that are mediated by these genes. The metabolism of gastric cancer is also affected by the regulation of many classical pathways, such as the hypoxia inducible factor (HIF-1a) pathway and the insulin signaling pathway (*Yuan, Yamashita & Seto, 2016*). The results showed that fatty acid synthase (FASN), a new metabolic reprogramming agent, also has prospective clinical applications. Metabolic reprogramming is emerging as a novel hallmark of cancer, and it is particularly important to identify biomarkers with high sensitivities and specificities at the metabolic level for the diagnosis and prognosis of STAD.

## MATERIALS AND METHODS

### Raw data

Transcriptome RNA sequencing (RNA-seq) data of 407 samples (normal samples, 32 patients; tumor samples, 375 patients) and the corresponding clinical data were downloaded from The Cancer Genome Atlas (TCGA) database (https://portal.gdc.cancer.gov/). The GSE84437 data with 433 tumor samples was downloaded from the GEO database (*Yoon et al., 2020*). The metabolic genes were retrieved from the metabolic pathways of GSEA (c2.cp.kegg.v7.1.symbols.gmt).

### Identification of intersected differentially expressed mRNAs in TCGA-STAD

The limma R software package (version 3.6.2) was used to analyze the differential expression of the annotated protein coding genes and the expression patterns of the 940 metabolic

genes were studied in TCGA. The 940 metabolic genes were selected as being consistently altered metabolic genes for subsequent prognostic analyses in the GSE84437 datasets. Metabolism-related genes shared by TCGA and GSE84437 datasets.

### Construction of the prognostic metabolic gene signature

The prognosis-related metabolic genes were confirmed by combining the univariate Cox regression and LASSO penalized Cox regression analysis and to set up a new prognostic gene signature. $P < 0.001$ was selected as the screening condition in univariate regression analysis. The prognostic gene signature was calculated by (coefficient$_{mRNA1}$ ×expression of mRNA1) + (coefficient$_{mRNA2}$ × expression of mRNA2) + (coefficient$_{mRNAn}$ × expression of mRNAn). The optimal cutoff point of the above prognostic gene signature and the Kaplan–Meier survival curve were conducted by R package (survival, survminer). The predictive performance was shown by ROC curve. Multivariate Cox regression analysis was conducted by forward stepwise analysis method and a nomogram was conducted based all of the independent prognostic factors.

### External validation of the prognostic gene signature and gene changes in the GEO data

The GSE84437 data set was included to calculate the risk score of the included patients with the gene signature. ROC and Kaplan–Meier analyses, as well as the construction and validation of the nomogram, were performed identically as those analyses in the cohort TCGA-STAD.

### Gene set enrichment analyses

Hallmark sets v 6.2 collections were downloaded from the Molecular Signatures Database as the target sets with which GSEA was performed by using GSEA 3.0 software. The entire transcriptome of all of the tumor samples was used for the GSEA, and only gene sets with NOM $P < 0.05$ and FDR $Q < 0.05$ were considered to be statistically significant.

## RESULTS

### Construction of the prognostic metabolic gene signature in TCGA

There are 192 metabolic genes in the TCGA-STAD database, including 119 up-regulated genes and 73 down-regulated genes (Fig. 1). 16 survival-related genes were confirmed by a univariate Cox regression analysis; then 14 survival-related genes were identified by LASSO-penalized Cox analysis; finally, a prognostic model was constructed based on 14 survival-related genes. The 14 genes included acylphosphatase 1 (ACYP1), RNA polymerase I subunit A (POLR1A), retinol dehydrogenase 12 (RDH12), 5-oxoprolinase, ATP-hydrolyzing (OPLAH), malic enzyme 1 (ME1), nicotinamide N-methyltransferase (NNMT), gamma-glutamyl transferase 5 (GGT5), deoxycytidine kinase (DCK), galactosidase alpha (GLA), DNA polymerase delta 3 (POLD3), glutathione S-transferase alpha 2 (GSTA2), N-acyl sphingosine amidohydrolase 1 (ASAH1), and N-acyl sphingosine amidohydrolase 1 (CKMT2). The risk score = 0. 0258×expression of ME1 -0. 0458×expression of ACYP1 -0. 0583×expression of POLR1A + 0. 0097×expression of RDH12 -0. 0109×expression of OPLAH + 0. 0039×expression of NNMT +0. 009411×expression of GGT5 - 0.

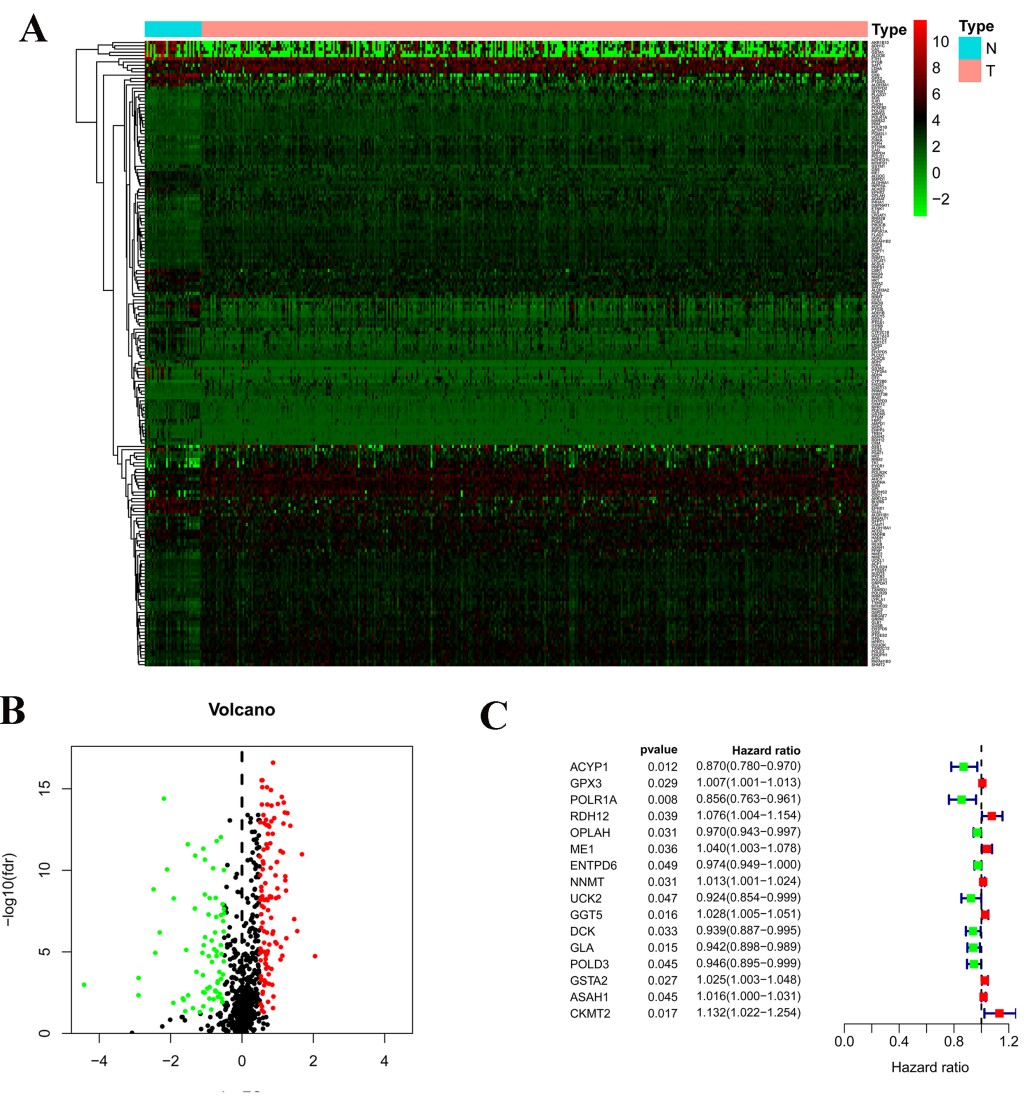

**Figure 1** **Construction of the prognostic metabolic gene signature in TCGA.** (A) Heatmap for expression level between normal patients and tumor patients; (B) Volcano for expression level between normal patients and tumor patients; (C) Forest map for univariate COX regression analysis with 192 different expression genes (DEGs), with the top 16 being listed.

0097×expression of DCK - 0. 0372×expression of GLA-0. 0062×expression of POLD3 + 0. 01629×expression of GSTA2 +0.0128 expression of ASAH1 + 0.0841 expression of CKMT2.

## Validation of the prognostic metabolic gene signatures in TCGA and GEO

According to the median risk score, patients were divided into high- and low-risk groups. The overall survival (OS) was significantly poorer in the high-risk group than in the low-risk group ($P < 0.0001$; Fig. 2A). As shown in Fig. 3C, people who died were more obviously distributed on the outlier risk scores than near the median risk scores. Subsequently, the
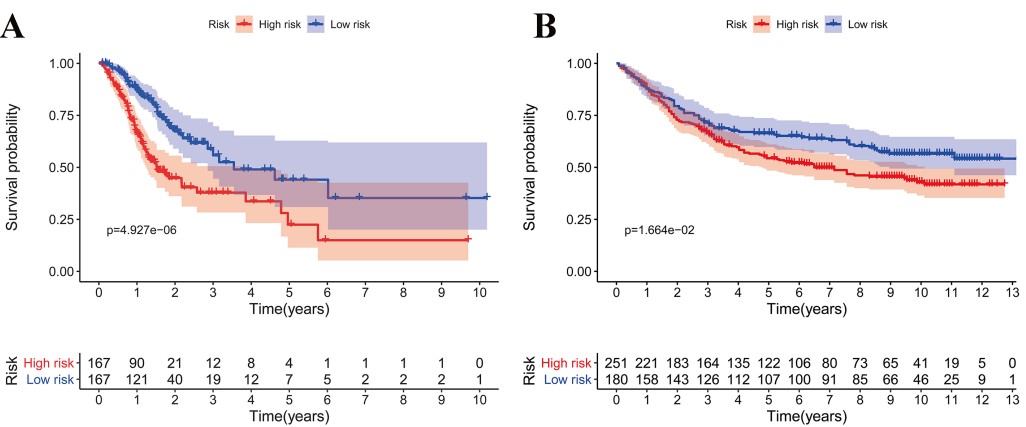

**Figure 2  Survival analysis of prognostic metabolic gene signatures in TCGA and GEO.** (A) Kaplan–Meier curve of the four-gene signature in TCGA cohort; (B) Kaplan–Meier curve of the four-gene signature in GSE84437.

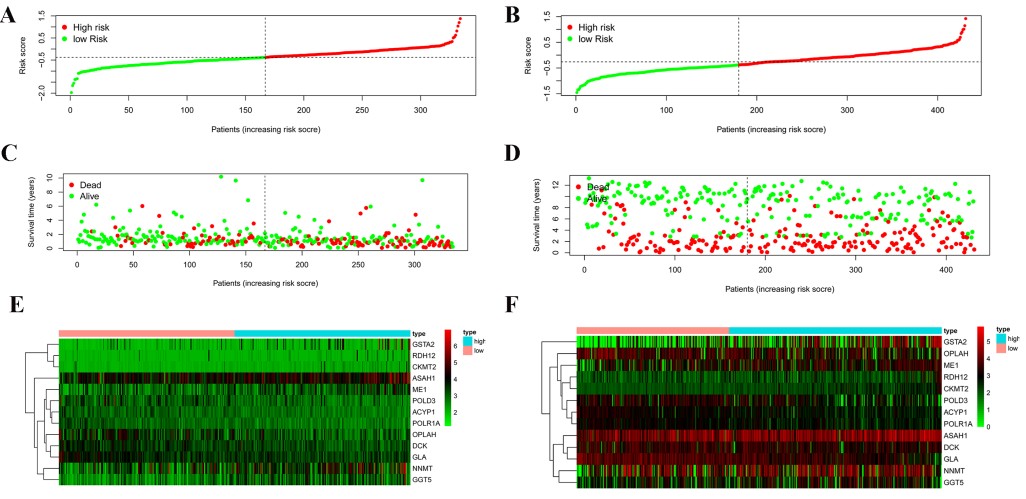

**Figure 3  Validation of the prognostic metabolic gene signatures in the TCGA and GEO.** (A, B) Risk scores of patients with different metabolic gene signatures in TCGA and GEO; (C, D) Survival states distribution of patients with different metabolic gene signatures in TCGA and GEO; (E, F) Heatmap for Expression level of prognostic metabolic gene.

prognostic model was validated in the GSE84437 cohort. According to the median risk score, patients were divided into a high- risk and low-risk group. The OS was significantly poorer in the high-risk group than in the low-risk group ($P < 0.0001$; Fig. 2B). As shown in Fig. 3D, people who died were more distributed on the outlier risk scores than near the median risk score. The expression of GSTA2, ME1, and CKMT2 in the high-risk group were higher than those in the low-risk group; however, the expression of OPLAH and GLA in the high-risk group were lower than those in the low-risk group (Fig. 3F).

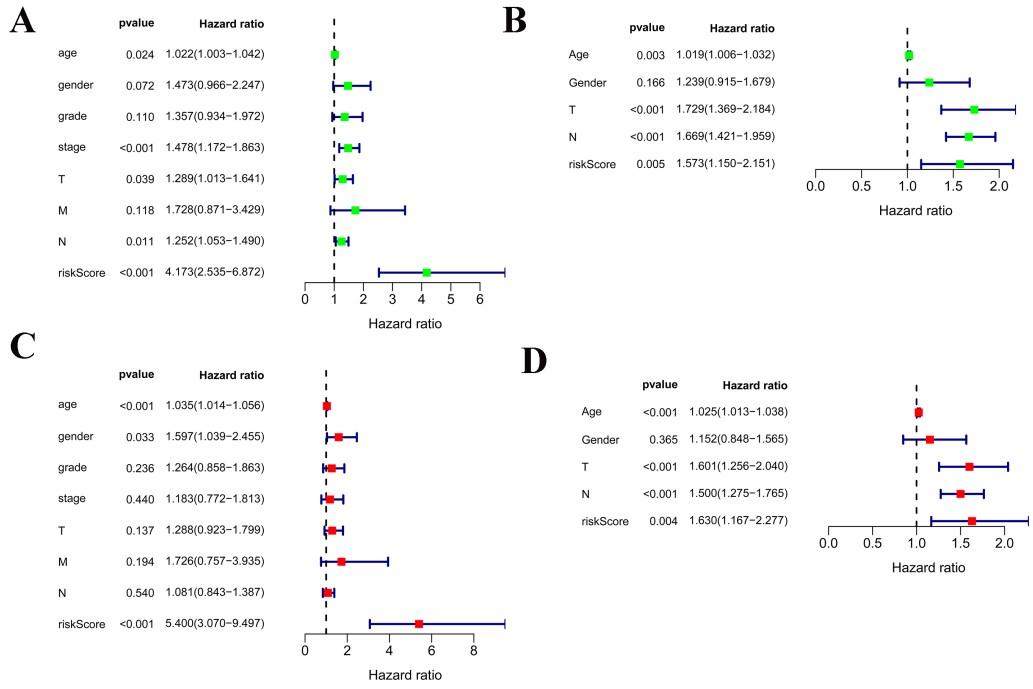

**Figure 4 Independent prognostic role of the prognostic gene signatures.** (A, B) Univariate Cox regression analysis for clinical characteristics and prognostic gene signatures in TCGA and GEO; (C, D) Multiple Cox regression analysis for the clinical characteristics and prognostic gene signature in TCGA and GEO.

## Independent prognostic role of the prognostic gene signature

Among the 375 patients who were included in the TCGA-STAD cohort, univariate and multivariate Cox regression analyses indicated that age and risk score were both independent prognostic factors for OS (Figs. 4A, 4C). Importantly, our risk scores were also independent prognostic factors for OS via the analysis of the 433 patients who were included in the GSE84437 cohort, which was consistent with the results from the cohort TCGA-STAD (Figs. 4B, 4D).

As shown in Fig. 5A, the area under the ROC curve (AUROC) of the risk score (AUROC = 0.696, sensitivity = 56.45%; specificity = 74.29%, Youden index = 0.307) was higher than that of the other parameters. In the GSE84437 cohort, the performance analysis of the discriminative accuracy of the risk score for mortality had an AUROC of 0.574 (sensitivity = 54.12%; specificity = 63.49%, Youden index = 0.176), which was also significant (Fig. 5B). A nomogram was built by including the TNM stage and the prognostic model (Fig. 6A) in the TCGA-STAD cohort (Fig. 6A). Age and risk score were determined to be the best parallel parameters for prognosis. Importantly, the risk score was also the best parallel parameter for prognosis in the GSE84437 cohort. Therefore, our prognostic model may have the potential to be a marker for STAD prognosis, which may help with the clinical management of STAD.

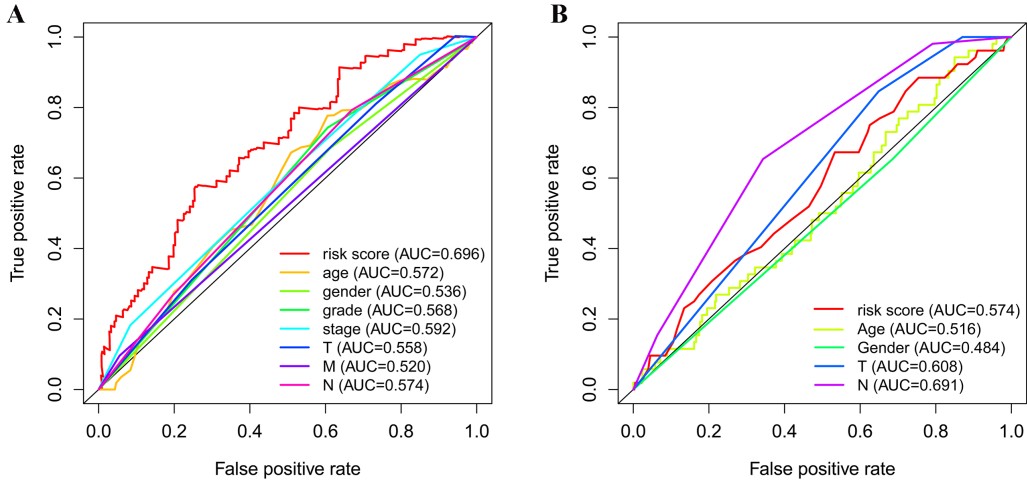

**Figure 5** Receiver operating characteristic curves of the clinical characteristics and prognostic gene signatures. (A) ROC for TCGA; (B) ROC for GEO.

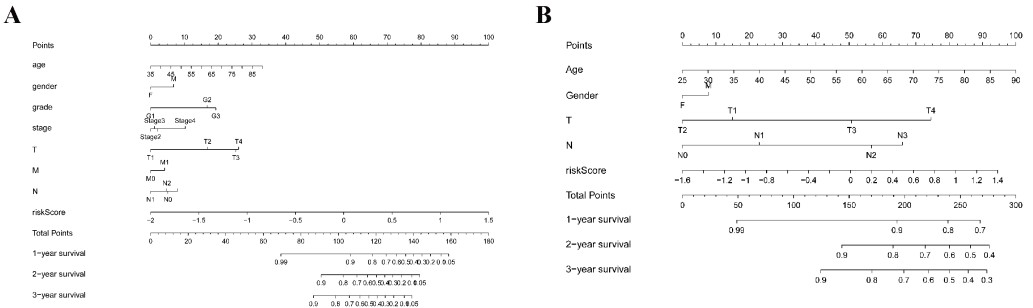

**Figure 6** Nomogram plot for the clinical characteristics and prognostic gene signatures. (A) Nomogram plot for TCGA; (B) Nomogram plot for GEO.

## Gene set enrichment analyses

GSEAs were performed, and 17 significantly enriched Kyoto Encyclopedia of Genes and Genomes (KEGG) pathways were found in the TCGA-STAD or GSE84437 cohort. As shown in Fig. 7B, many of the enriched pathways were related to metabolism, such as drug metabolism of P450, arachidonic acid metabolism, retinol metabolism, and pyrimidine metabolism. In addition, most of the metabolism-related pathways were enriched in the low-risk group, whereas most of the pathways that were not related to metabolism were enriched in the high-risk group.

## DISCUSSION

Early diagnosis is the key to improving the survival rates of patients. In recent years, a large number of studies have shown that molecular biomarkers play an important role in disease diagnosis, prognosis prediction and targeted therapy. STAD is one of the most serious malignant tumors in the world, with a high mortality rate and a poor prognosis.

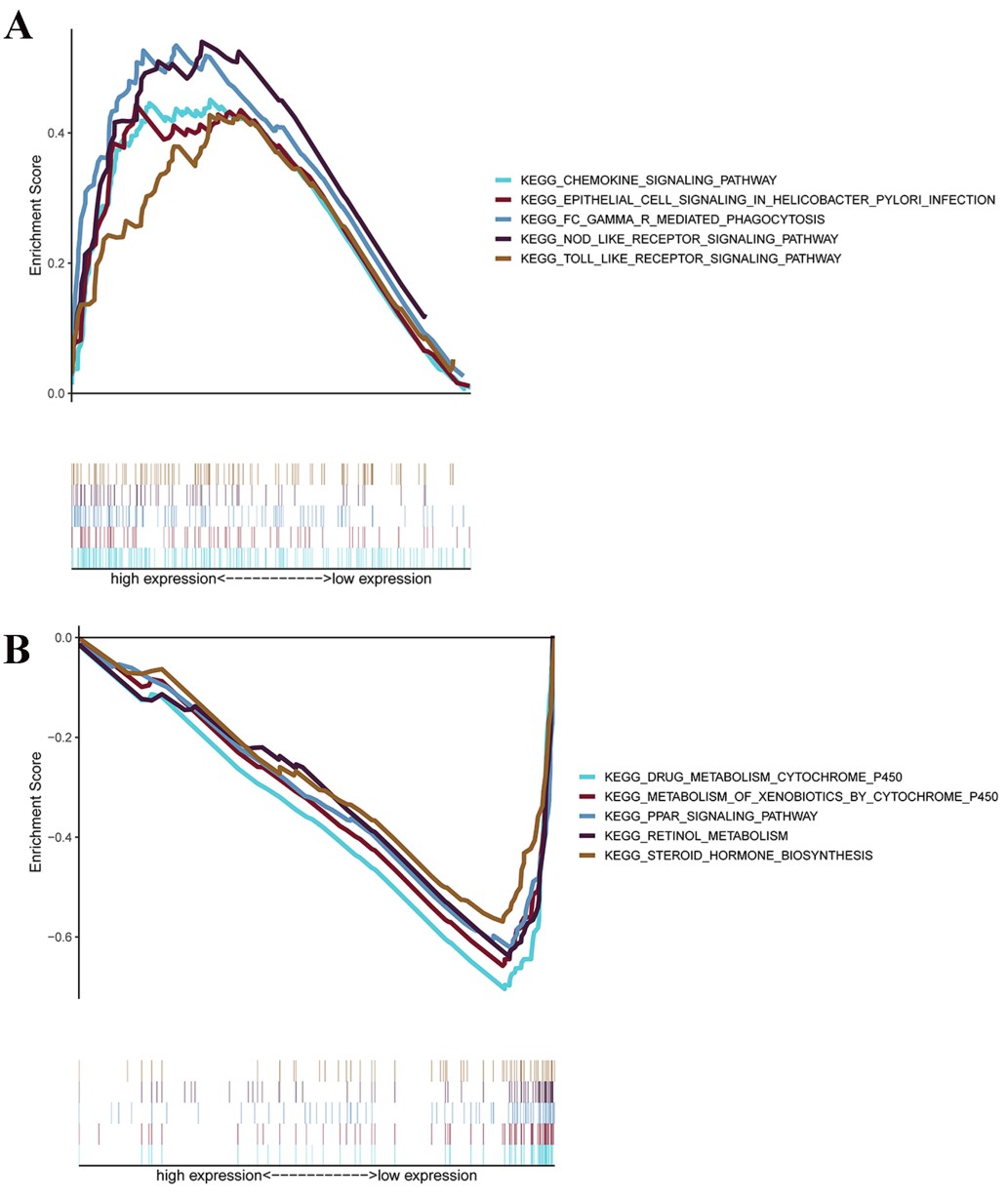

**Figure 7** **GSEA for samples with high levels of prognostic gene signature and low expression.** (A) GSEA for samples with high expression levels; (B) GSEA for samples with low expression levels.

*Helicobacter pylori* infection, improper diet, poor hygiene and smoking are common risk factors for STAD (*Yang et al., 2011*). In addition, the delay in diagnosis and the metastasis of gastric cancer are the main causes of death for patients with gastric cancer. Therefore, the search for new molecular markers is very important for early diagnosis, targeted treatment and prognosis evaluation of gastric cancer (*Salati et al., 2019*).

Cancer is a metabolic disease. Metabolic disorder is a key event in the occurrence and development of cancer, and it constitutes one of the signs of cancer (*Faubert, Solmonson &*

*De Berardinis, 2020*). In a cohort study of 125 gastric cancer samples with different stages (I-N), 48 different metabolites were identified, 13 of which involved glycolysis, glutamine metabolism, amino acid metabolism, and choline metabolism, and these metabolites were related to the progression of gastric cancer, with a potential for staging diagnosis (*Wang et al., 2016*). The combined survival analysis of the serum metabolome of 125 gastric cancer patients showed that the serum levels of 2,4-hexadienoic acid, 4-tolyl dodecanoate and tributyrin were inversely related to the survival rates of patients, then suggesting that the combination of 3 serum metabolites may be an independent prognostic factor for gastric cancer (*Wang et al., 2017*). The plasma amino acid metabolism profiles of 82 patients with gastric ulcers and 84 patients with gastric cancer were compared. Five different amino acids (glutamine, ornithine, histidine, arginine, and tryptophan) showed good differentiation ability between gastric ulcer and gastric cancer (*Jing et al., 2018*). In summary, changes in metabolites can effectively predict the progression and prognosis of gastric cancer patients.

In recent years, mRNA gene signatures based on certain characteristics, such as long noncoding RNA have become a hot topic in research for mortality risk prediction in cancer (*Li et al., 2016*). In the study by Liu, GM, it was demonstrated that a four-gene metabolic signature has a predictive value in the OS for patients with hepatocellular carcinoma (*Liu et al., 2020a*); however, reports on the prediction of metabolism-related genes in gastric cancer are very limited. In this study, we identified a novel efficient metabolic prognostic signature based on the data set from TCGA and validated its efficiency in the GSE84437 data set. Our signature could efficiently stratify the OS values of patients. Via univariate and multivariate Cox regression analyses, the efficacy of our signature was found in the training set and in the validation set, thus indicating a robustly high prognostic value of the signature. In the GSEA cohort, most of the pathways in the high-risk group were mainly enriched in pathways that were not related to metabolism; however, most of the pathways in the low-risk group were mainly enriched in metabolism-related pathways.

Fourteen genes (ME1, ACYP1, POLR1A, RDH12, OPLAH, NNMT, GGT5, DCK, GLA, POLD3, GSTA2, ASAH1, and CKMT2) were involved in the disorder. In the study of acute myeloid leukemia, DCK is the rate-limiting enzyme for the metabolism of cytarabine after entering the cell, and changes in the properties of DCK directly affect the effective concentration of cytarabine (*Shi et al., 2004*). POLD participates in mediating the process of DNA amplification, replication and damage repair by interacting with proliferating cell nuclear antigen (PCNA) (*Zhou et al., 2018*). *Rayner et al. (2016)* reported that mutations in the POLD3 gene line can increase the risk of rectal cancer. Glutathione S-transferase (GST) is a very important enzyme superfamily in vivo that is involved in biotransformation and the detoxification process of many carcinogens. GST can form a DNA adduct after exposure to a pre-carcinogen, thus resulting in a high level of DNA damage. This results in the ineffective metabolism of the corresponding carcinogens, thus resulting in the accumulation of carcinogens in the body, which increases the risk of cancer. Mitochondrial creatine kinase 2 (CKMT2) is an important kinase that exists on the surface of the mitochondrial membrane and is directly related to intracellular energy transfer and ATP regeneration (*Cannavo et al., 2018*). CKMT2 is positively correlated with the malignant degree of gastric cancer. ASAH1 is a key enzyme that regulates the hydrolysis of intracellular

ceramide and plays an important role in cellular proliferation and apoptosis (*Roh et al., 2016*). The expression of ASAH1 in tumor tissues is positively correlated with breast cancer tumor size. There are many reports concerning the previously described genes, but the confirmed mechanisms of actions of these genes need to be further studied, especially in relation to gastric cancer.

There were some reports on the bioinformatics analysis of gene expression and the predictive prognosis of STAD, especially after the application of machine learning techniques in bioinformatics (*Quoc et al., 2020*; *Le et al., 2020*). From the perspective of alternative splicing, Liu et al. reported that 2,042 alternative splicing genes play an important role in regulating gastric cancer-related processes, such as GTPase activity and the PI3K-Akt signaling pathway, and they found that ECT2 may be a biomarker for diagnosis and prognosis (*Liu et al., 2020b*). The occurrence and prognosis of STAD are closely related to inflammation. Additionally, a prognostic model based on seven immune-related genes was developed (*Wu et al., 2020*). Metabolic recombination is an important characteristic of cancer, and glycolysis is an important part of this process. A gene signature based on a seven-gene signature of glycolysis was conducted, which has good calibration and moderate discrimination (*Yu et al., 2020*). *Zhao et al. (2020)* reported that BicC family RNA-binding protein 1 (BICC1) may be a potential prognostic biomarker in STAD and correlates with immune infiltrates. However, a bioinformatics analysis based on all of the metabolic genes is limited.

This study had several limitations. Firstly, although it has been verified by the GSE84437 cohort, the main aim of this study is represented by the bioinformatics analysis based on TCGA, and functional experiments are necessary to reveal the predictive mechanisms. Secondly, confounding effects of treatment factors are difficult to control because of the lack of treatment information and and it was difficult to reduce the batch effect. Lastly, the predictive performance of GSE84437 was not very good, which was related with the different pathogenesis mechanisms of stomach adenocarcinoma from different regions; thus, a larger, multicenter cohort is required. Finally, the AUROC between TCGA and GEO data was relatively large and may be reduced by enlarging the sample size, constructing convolutional neural networks (CNN) or the use of a support vector machine (SVM).

In conclusion, our study showed that a novel metabolic signature based on TCGA has the potential to be a prognostic factor for STAD patients. Our signature may reflect the dysregulated metabolic microenvironment and can provide potential biomarkers for metabolic therapy; however, validations of the signature and functional experiments are still needed.

### Funding

This study was supported by the National Natural Science Foundation of China (grant number: 81960120), "Gan-Po Talent 555" Project of Jiangxi Province (GCZ (2012)-1) and the Postgraduate Innovation Special Foundation of Jiangxi Province (YC2020-B046). The

funders had no role in study design, data collection and analysis, decision to publish, or preparation of the manuscript.

### Grant Disclosures

The following grant information was disclosed by the authors:

The National Natural Science Foundation of China: 81960120.

"Gan-Po Talent 555" Project of Jiangxi Province (GCZ (2012)-1).

The Postgraduate Innovation Special Foundation of Jiangxi Province (YC2020-B046).

### Competing Interests

The authors declare there are no competing interests.

### Author Contributions

- Yuan Nie conceived and designed the experiments, prepared figures and/or tables, and approved the final draft.
- Linxiang Liu performed the experiments, analyzed the data, prepared figures and/or tables, and approved the final draft.
- Qi Liu performed the experiments, prepared figures and/or tables, data collection, and approved the final draft.
- Xuan Zhu analyzed the data, authored or reviewed drafts of the paper, and approved the final draft.

### Data Availability

The source codes and collected data are available in the Supplemental Files.

### Supplemental Information

Supplemental information for this article can be found online at http://dx.doi.org/10.7717/peerj.10908#supplemental-information.

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
