# Peer review of "Identification of a metabolic-related gene signature predicting the overall survival for patients with stomach adenocarcinoma"

_PeerJ, doi:10.7717/peerj.10908_

## Round 0.1 · original submission · Major Revisions

Dear Dr. Zhu,
Thank you for your submission to PeerJ.

The review comments are now in hand. A few major concerns have been raised by the reviewers and as per my opinion these points are very important.

Therefore, please revise the paper as per the suggestions of the reviewers and resubmit for further processing.

Thank you,
Best regards,
Debmalya Barh, PhD

Reviewer 1 ·

Basic reporting

There are some minor grammatical errors, such as the lack of "of" in "expression mRNAn" on line 92, etc.

Experimental design

no comment

Validity of the findings

In the GEO cohort, compared with other clinical indicators, the "AUC" of "the risk score" is not very good. The author should discuss the possible reasons.

Additional comments

GSEA is used to analyze all possible KEGGs of genes in the model. It is recommended that the author add GO and KEGG analysis to the differential genes of "the high-risk group" and "the low-risk group".

Reviewer 2 ·

Basic reporting

The manuscript by Yuan Nie et al. demonstrates novel prognostic markers which help in early diagnosis in Stomach adenocarcinoma (STAD). Using databases from TCGA authors have established prognostic model systems and validated by GSE84437 cohort. Manuscript is properly referenced and data is professionally tabulated. However, manuscript have grammatical and spelling mistakes at places. It would be useful if manuscript can be prrofread by a native speaker before resubmission

Experimental design

1. Authors have concluded their entire findings based on TCGA and GEO databases only. It would have been more tempting to include other cancer data repository e.g Oncomine ,Bioportal etc.

2. In addition to univariate Cox regression author can also use survival trees (e.g.CART for survival data) analysis which will further support the finding.

Validity of the findings

No comments

Additional comments

The manuscript by Yuan Nie et al. demonstrates novel prognostic markers which help in early diagnosis in Stomach adenocarcinoma (STAD). Using databases from TCGA authors have established prognostic model systems and validated by GSE84437 cohort. I find manuscript can be considered for publication in PeerJ with addressing a few following minor comments.

1. The authors have concluded their entire findings based on TCGA and GEO databases only. It would have been more tempting to include other cancer data repository e.g Oncomine ,Bioportal etc.

2. In addition to univariate Cox regression author can also use survival trees (e.g.CART for survival data) analysis which will further support the finding.

3. Manuscript have grammatical and spelling mistakes at places. It would be useful if manuscript can be prrofread by a native speaker before resubmission.

Reviewer 3 ·

Basic reporting

The authors proposed a bioinformatics study to identify a metabolic-related gene signature
predicting the overall survival of patients with stomach adenocarcinoma. Some efforts have been done, however, there are some major points that need to be improved:

- The use of English should be improved significantly. There are some grammatical errors and typos. It is better to be checked by a native speaker or an English editing service.

- Lacking literature review. The authors missed to mention and discuss a lot of related bioinformatics studies on stomach adenocarcinoma specifically.

- Quality of figures should be improved.

Experimental design

- Did the authors concern about the batch effect removal in their study?

- TCGA & GEO have been used in previous bioinformatics studies such as https://doi.org/10.1016/j.meomic.2020.100001 and https://doi.org/10.3390/jpm10030128. Therefore, the authors should mention more works to attract a broader readership.

- The authors should release source codes for replicating the methods.

- Why did the authors select median values as the cut-off for risk score?

Validity of the findings

- Performance results were not so good (about AUC of 0.57 in validation set)

- I cannot see the legends in the heatmap, it is important to improve the quality of figures.

- How did the authors calculate the risk-scores for four-gene signatures together? As I have known, it is easier to calculate risk-score for individual genes.

- ROC curve or AUC has been used in previous bioinformatics studies i.e., PMID: 31362508 and PMID: 31921391. It is suggested to refer to more works in this description.

- In Fig. 5, it is better to use the same color for the same characteristic among TCGA and GEO set. It is not easy to compare with different color use.

- In Fig. 5 also, why did GEO dataset have fewer characteristics (fewer lines) than TCGA?

- There is overfitting on the results (AUC of 0.696 and 0.574 in training and validation set). The authors should discuss how to deal with it.

Additional comments

No comment

---

## Round 0.2 · accepted · Accept

Dear Dr. Zhu,

Happy New Year!

All the three reviewers have recommended and it is my pleasure to accept your revised article.

Thank you,
Best regards,
Debmalya Barh, PhD

Reviewer 1 ·

Basic reporting

No comment.

Experimental design

No comment.

Validity of the findings

No comment.

Additional comments

No comment.

Reviewer 2 ·

Basic reporting

No comments

Experimental design

No comments

Validity of the findings

No comments

Additional comments

Authors have provided a logical and satisfactory point-by-point explanation for the concerns raised during review process. I think that manuscript has improved substantially and can be considered for publication.

Reviewer 3 ·

Basic reporting

no comment

Experimental design

no comment

Validity of the findings

no comment

Additional comments

My previous comments have been addressed satisfactorily.